# Development of a Global Health Learning Progression (GHELP) Model

**DOI:** 10.3390/pharmacy9010002

**Published:** 2020-12-24

**Authors:** Ellen M. Schellhase, Monica L. Miller, Jodie V. Malhotra, Sarah A. Dascanio, Jacqueline E. McLaughlin, David R. Steeb

**Affiliations:** 1College of Pharmacy, Purdue University, Indianapolis, IN 46202, USA; mille355@purdue.edu; 2School of Pharmacy and Pharmaceutical Sciences, University of Colorado, Aurora, CO 80045, USA; jodie.malhotra@cuanschutz.edu; 3UNC Eshelman School of Pharmacy, University of North Carolina at Chapel Hill, Chapel Hill, NC 27599, USA; saraharvilla@gmail.com (S.A.D.); jacqui_mclaughlin@unc.edu (J.E.M.); david_steeb@unc.edu (D.R.S.)

**Keywords:** global health, study abroad, learning outcomes, experiential education, educational models

## Abstract

There has been a steady increase in global health experiential opportunities offered within healthcare professional training programs and with this, a need to describe the process for learning. This article describes a model to contextualize global health learning for students who complete international advanced pharmacy practice experiences (APPEs). Students from University of North Carolina at Chapel Hill, Purdue University, and the University of Colorado completed a post-APPE survey which included open-ended questions about knowledge, skills, and attitudes one week after completing an international APPE. Students were also invited to participate in a focus group. All 81 students who participated in an international APPE completed the open-ended survey questions and 22 students participated in a focus group discussion. Qualitative data from both the survey and focus groups were coded in a two-cycle open coding process. Code mapping and analytic memo writing were analyzed to derive to a conceptual learning model. The Global Health Experience Learning Progression (GHELP) model was derived to describe the process of student learning while on global health experiences. This progression model has three constructs and incorporates learning from external and internal influences. The model describes how students can advance from cultural awareness to cultural sensitivity and describes how student pharmacists who participate in international experiential education develop global health knowledge, skills, and attitudes.

## 1. Introduction

With the growing level of globalization throughout the world, there have been increasing discussions about how to prepare college graduates for the global workforce. This conversation is also growing within healthcare professional training programs [1,2]. To address this need there has been a steady increase in global health experiential opportunities offered within healthcare professional training programs such as medicine and pharmacy [2,3,4,5]. Advanced pharmacy practice experiences (APPEs) are a common placement for global health experiences within curricula. Like other learning environments, there are established learning outcomes related to knowledge, skills, and attitudes for each experience. Preliminary outcomes and benefits of these opportunities for health professionals include increased cultural appreciation, identity formation, cross-cultural communication, and confidence [6,7].

While there have been documented benefits for global health experiential learning opportunities, there are limited frameworks to describe the process of learning that takes place to achieve these learning outcomes. Understanding how students learn in other disciplines has led to the development of key educational models, including Kolb’s Experiential Learning Cycle and Mezirow’s Transformative Learning Theory [8,9]. Kolb’s theory describes how a learner understands and processes information while Mezirow’s theory examines the use of critical self-reflection to foster change. These models have highlighted a need to create transformative, immersive experiences that allow the learner to experience a type of disorienting dilemma that enables critical reflection as to how one may change their thinking and attitudes [10,11,12]. While these models are useful for helping frame experiential learning, there is not a model developed for use in global health learning across health professions. 

In 2015, The Consortium of Universities for Global Health (CUGH) set out to develop a framework which was intended to be utilized as a starting point for “global citizen” global health competencies for all trainees across healthcare disciplines [13]. Given the recent development of these core competencies and their broad applicability, little research has been conducted on how these competencies are used to design or assess global health learning experiences. Further, there is no research on how the learning outcomes from these core competencies may interact with each other to explain how students actually learn while abroad. From 2017–2018, researchers from three Universities assessed knowledge, skills, and attitudes of student pharmacists who completed international APPEs using the CUGH global competency framework [14,15]. The purpose of this study is to describe the development of a new learning model that helps describe how student pharmacists acquired their learning outcomes during an international APPE.

## 2. Methods 

### 2.1. Program Description

Eighty-one final year student pharmacists from Purdue College of Pharmacy, the University of Colorado Skaggs School of Pharmacy, or the University of North Carolina at Chapel Hill Eshelman School of Pharmacy completed an international APPE during the 2017–2018 academic year. APPEs were completed in both low-middle income (LMICs) and high-income countries (HICs) and were 4 (n = 39), 6 (n = 12) or 8 (n = 30) weeks in duration. Countries included: China, Ethiopia, Guatemala, India, Kenya, Malawi, Moldova, Tanzania, and Zambia (LMICs (n = 48)), and Australia, Canada, Japan, New Zealand, and the United Kingdom (HICs (n = 33). Student pharmacists participated in a variety of experiences that included direct patient care and non-patient care activities. 

### 2.2. Data Collection

All student pharmacists completed a post-APPE self-assessment Qualtrics™ (Qualtrics, Provo, UT, USA) survey within one week of completing their APPE. As part of the survey, students answered open-ended questions targeted at the knowledge, skills, and attitudes gained or enhanced while on their international experience. Students received the open-ended questions one week prior to the survey distribution to allow for reflection prior to survey completion. In addition to the survey, all students were invited to participate in a virtual focus group via ZOOM 4.1 (Zoom Video Communications, San Jose, CA, USA) to provide further context regarding their experiences and how they impacted their global health learning outcomes. Focus group facilitators followed a script and questions were given to participants one week in advance to allow for reflection. A total of 22 students participated in the focus groups which were conducted at least one month after completing their APPE.

All qualitative data from completed surveys and focus groups were analyzed using MAXQDA 2018 (VERBI Software GmbH, Berlin, Germany). Investigators conducted a two-cycle, open coding process using conventional content analysis in which coding discrepancies were resolved through face-to-face meetings to form a codebook for each open-ended survey question and focus group response. The coding process identified 25 knowledge codes, 26 skills codes, and 19 attitude codes. The codebook, which includes all knowledge, skills and attitude codes, definitions, and representative student responses, is available in Appendix A. 

### 2.3. Study Methods

While the objective of two prior studies of this research was to determine *what* students learned while abroad on international APPEs, discussion with the research team and analysis of the analytic memos from the coding process revealed new questions as to *how* students learned these outcomes while abroad [14,15]. Utilizing the aforementioned codebook (Appendix A), the researchers went back to the original open response survey and focus group data and analyzed these through the lens of Mezirow’s Transformative Learning Theory and Kolb’s Experiential Learning Theory to examine the connectivity between codes as students progressed on their international APPE [8,9]. The research team focused on exploring the relation between the primary coded themes from both a temporal and cyclical perspective to explore learning patterns across country locations. Each team member reviewed the original data and recorded new analytic memos. The data were first reviewed by learning outcome category (knowledge, skills, and attitudes) for all students while the second review was by individual students across these learning outcomes. The team met several times thereafter to discuss the findings of the analytic memos and to categorize the original predominant codes into a conceptual learning model. From a temporal perspective, the predominant codes were labeled as either early, transitional, or later in accordance to whether they took place at the beginning, middle, or end of the APPE rotation experience. The learning model was then sent to two global health experts in pharmacy, not involved with the prior study, for face and content validity.

## 3. Results

Table 1 presents the predominant global health learning outcomes that resulted from the coding process for knowledge, skills, and attitudes while students were on their international experience. Knowledge-based learning outcomes were labeled as occurring early in the APPE, the skills-based learning outcomes took place in the middle and were considered transitional, while the attitude-based learning outcomes occurred later towards the end of the APPE. While the original coding process revealed what students were learning, further analysis of the responses, codes, and analytic memos revealed a process of learning that helped explain how students developed the discovered learning outcomes. This process is presented in Figure 1 as the Global Health Experience Learning Progression (GHELP) model. The two external expert reviewers concurred with the model design, indicating that it would be applicable to global health experiences at their home institution, and had no substantive feedback. 

Both written comments from the survey and verbal comments from the focus groups revealed how students initially seemed overwhelmed with how to conceptualize their new environment in terms that made sense to them. They indicated that vast cultural and patient care differences led them to apply communication, problem solving, and adaptability skills to develop an understanding of these differences and navigate new cultural norms. As students utilized these skills with patients and practitioners to understand why things are different, students further enhanced their cultural awareness. A smaller proportion of students went on to realize that they could succeed and thrive in their new environment, which coincided with an increase in self-awareness and self-efficacy described as confidence. 


*“I think the biggest impact this rotation will have on my future career is the confidence that I gained by putting myself in a completely new environment and thriving. I have always had some connection in one way or another and never truly had the “outside looking in” experience. This rotation has provided me that experience and has pushed my comfort zone many times, but I have been able to adapt as needed. It gives me confidence that no matter where I end up after graduation, I have the skills needed to assimilate to the environment.”*
—*Participant 15 (survey)*

This shift in mentality often resulted in students becoming more appreciative of the differences they had to navigate and introspective as to how new cultural beliefs and values compared against their own. Students commented on how the experience transformed their personal and professional outlook, often giving them new insight into what global health was about and motivating them to become a better practitioner to improve patient care. Students often mentioned a newfound respect for alternative approaches to patient care, noting that just because something is different, does not mean it is wrong. Students became more sensitive to some of the cultural differences, from language barriers to health behaviors, and wanted to apply what they learned back home.


*“From this experience I have learned how to approach issues with a more open mindset. Instead of thinking of the practices (cultural and medical) in Malawi as being “wrong”, I learned to ask why things were done differently. Continuing this practice in the future will make me a more empathetic and understanding provider.”*
—*Participant 18 (survey)*

## 4. Discussion

### 4.1. GHELP and Other Models for Experiential Learning

The GHELP model is an outcome of the original data analysis of two prior studies [14,15] that helps explain how students’ knowledge, skills, and attitudes progress over time while on global health experiences. This progression model has three constructs (cultural awareness, appreciation and sensitivity) and incorporates learning from external and internal influences. A review of the literature revealed other learning models addressing experiential and/or cultural learning. When evaluating other published learning models that may align with our findings, there were identified similarities but none that focused on experiential learning, cultural competence, and global health training outcomes for health professionals. Table 2 provides a comparison of other models with the GHELP model [16,17,18,19,20,21,22].

Two models in particular correlate with the GHELP model and the learning progression students make during global health experiences. The first model that correlates with the learning process our students went through was Kolb’s Experiential Learning Theory [8]. Both Kolb’s and GHELP are rooted in experiential learning and both highlight how students go through the process of experiencing, thinking, reflecting, and acting for effective learning [8]. Additionally, the GHELP model highlights how students use reflective observation to conceptualize the differences they encounter and experiment with how to apply their newfound insight which mirrors Kolb’s theory [8]. The student pharmacists within this study also showcased a process for how they gained confidence while navigating unfamiliar situations. It was noted, they spent less time figuring out differences and more time reflecting on the meaning behind differences, which led to an appreciation for their new cultural environment. The shift from cultural awareness to cultural appreciation and sensitivity through critical reflection is similar to the cultural progression seen as one starts to go from the cognitive to the affective domain of learning [22].

Mezirow’s Transformative Learning Theory is the second learning model that has similarities to the GHELP model [9]. Mezirow’s theory states that transformation is a ten phase process beginning with facing a disorienting dilemma that then stimulates self-reflection leading to the exploration of new roles and perspectives [9]. As one progresses through the ten phases, they must critically reflect on abstract scenarios and make meaning of their reflections in order to actively experiment and test their understanding. This process of translating observations into meaning is seen in GHELP as students utilize the skills of communication, problem solving, and adaptability to understand the disorienting dilemma of the new culture. Once students better understand their external environment, they use these same skills in an introspective manner as they begin to go through the process of negotiating their identity both personally and professionally. International experiences allow students to go through components of Mezirow’s process of transformative learning and can enable professional identity formation to help students define their purpose within the profession [23]. Student experimentation with new identity roles and increased confidence in these roles can be applied locally, back home, which is similar to Mezirow’s last phase of reintegrating new perspective in one’s life. As disorienting dilemmas can happen in a variety of settings locally and globally, exploration of the utility and application of the GHELP model can be considered in other contexts beyond just international locations [24,25,26].

### 4.2. GHELP and Other Models for Cultural Competency Learning

There are many different terms used when describing “culture”. Competence, intelligence, and sensitivity are a few of these terms. In many cases there is ambiguity within the literature around these definitions that can lead to confusion about learning outcomes related to cultural growth [27,28]. The GHELP model contains elements of different cultural competence models, yet the progression from awareness to sensitivity is unique.

The Four Factor Model for Cultural Intelligence™ developed by Van Dyne and colleagues uses a four-factor framework including motivational, cognitive, metacognitive, and behavioral constructs [29]. The GHELP model has several common elements from the constructs in this model but is focused on healthcare learning (with clinical and non-clinical learning) and demonstrates a progression starting with cultural awareness. The origin of Van Dyne’s model was for the workplace and the scope of the model has expanded to include study abroad experiences but is not discipline specific, whereas the development of the GHELP is rooted in a healthcare context.

The Papadopoulos, Tilki and Taylor Model for Developing Cultural Competence was originally developed to address cultural learning in nursing students [30]. It includes four stages: awareness, knowledge, sensitivity, and competence. The GHELP model does not include knowledge as a stage but instead focuses on students’ experiencing, thinking, reflecting, and acting as part of the progression. While there is some overlap in the terminology used to describe the stages of development within each model, the two models do not directly align.

The Pyramid Model of Intercultural Competence builds on one’s knowledge, skills, and attitudes using attitudes as the foundation for the development of intercultural competence [18]. This model has some correlation with the GHELP model as it addresses knowledge, skills, and attitudes, which were assessed in the development of the GHELP model, however, it does not specifically address healthcare. While the GHELP model uses external and internal factors to drive progress from awareness toward cultural sensitivity, the pyramid model climbs to a peak with the top of the pyramid being external outcomes.

Another model for cultural competence that can be compared with the GHELP model is The Process of Cultural Competence in the Delivery of Healthcare Services developed by Campinha [16]. Similar to the GHELP model it is focused on the development of cultural competence in healthcare. This model has four constructs (awareness, knowledge, skill, and encounter) and is described as a process however there is not a focus on how a learner progresses through the model, as seen in the GHELP model. There are three versions of the model with the latter two adding “cultural desire” as a motivator. Within the GHELP model, motivation is incorporated as a way students apply their cultural sensitivity in a local context.

### 4.3. Limitations

There are several limitations to consider when evaluating the qualitative data used to inform the model [31,32]. Data came from student pharmacists who had varied international experiences. Variations included locations, length of experience, and rotation activities. While these may have provided data from diverse international experiences, it may also have impacted the survey responses and predominant learning outcomes that were coded. Data were also collected retrospectively. Timing of the survey and focus groups could have impacted student responses. These differences may also have impacted the prevalence of the different codes identified. In addition, there could have been social desirability bias in the responses. To limit this bias, the survey was anonymous and leading questions were avoided in both the survey and the focus groups.

### 4.4. Additional Assessment and Utility of GHELP

The GHELP model can serve as a resource to better structure and design global health experiences to maximize transformational learning and impact. It is possible that the degree of student transformation from the experience may correlate with the relative background of the student compared to where the student is being placed. Study abroad literature has demonstrated that those students who come from diverse cultural backgrounds have less of an impact from study abroad experiences compared to those with less diverse backgrounds [33,34,35]. Students from these diverse backgrounds may interpret disorienting experiences as more familiar and hence not have as high of a level of progression within the model. Additionally, the GHELP model can serve as a framework for reflective debriefs to help students reinforce the identified learning outcomes and further progress towards cultural sensitivity.

The GHELP model is a theoretical model rooted in pharmacy experiential education that should be assessed and validated to determine if participation in global health experiential learning allows students to move through this process. Further validation of the applicability of the model within pharmacy, as well as other healthcare disciplines should be considered. Assessment should include identifying if there are specific activities that better help students move from cultural awareness to appreciation to sensitivity. There may also be applicability to students participating in global health study abroad programs that have both clinical and non-clinical experiences. Another consideration is the use of this model in pre-departure training for global health experiential learning as well as a framework for reflective debriefing, perhaps better instilling a sense of identity as a global citizen and shifting to a global mindset. Awareness of this learning model may help students increase ownership of their learning process and enhance their transformation.

## 5. Conclusions

A global health learning progression model emerged from qualitative analysis, illustrating the utilization of new knowledge and skills to make meaning of cultural and patient care differences. Disorienting experiences coupled with critical self-reflection allows for progression through the GHELP model. While there are models to explain learning in experiential or study abroad settings as well as models for cultural competence, the GHELP model adds something new to the literature. This model showcases how healthcare students, specifically student pharmacists, develop a global mindset by acquiring the knowledge, skills and attitudes needed to achieve the global health experiential learning outcomes.

## Figures and Tables

**Figure 1 pharmacy-09-00002-f001:**
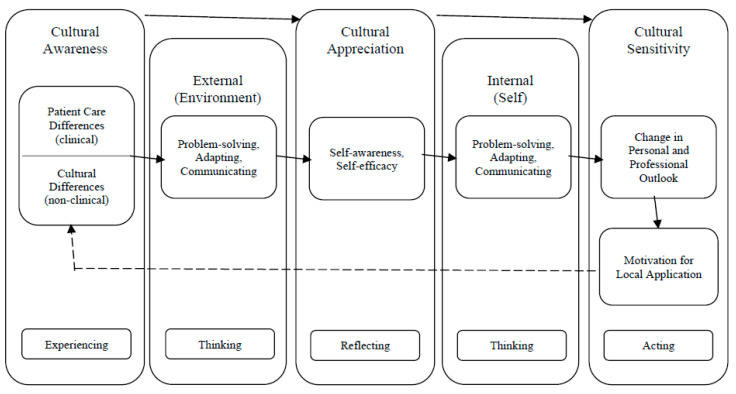
Global health experience learning progression (GHELP) model. This model explains the cultural progression and learning outcomes of students while on global health experiences. Cultural and patient care differences trigger exploration within their external environment leading some to progress from cultural awareness to cultural appreciation. As students reflect on these differences and question their own values and beliefs, some students progress to cultural sensitivity in wanting to incorporate their newfound knowledge into their future practice. This process can lead to a new cultural baseline, motivating the student to address the next patient care or cultural differences from a new perspective. This cyclical process incorporates elements of Kolb’s Experiential Learning Theory and Mezirow’s Transformative Learning Theory [8,9].

**Table 1 pharmacy-09-00002-t001:** Global health learning outcomes from coding process.

	Learning Outcome
Knowledge	Cultural awareness Patient care differences
Skills	Adaptability Problem solving Communication Self-awareness Self-efficacy
Attitudes	Cultural appreciation Cultural sensitivity Personal and professional outlook Global health perspective

**Table 2 pharmacy-09-00002-t002:** Example models.

Model	Healthcare Specific	# of Constructs	Type of Model
**Experiential Learning Models**			
Kolb’s Experiential Learning Cycle (Kolb, 2000)	No	4 (cycles)	stages
Mezirow’s Theory of Transformational Learning (Mezirow, 1990)	No	10 (phases)	progression
**Cultural Learning Models**			
Four Factor Model for Cultural Intelligence (CQ Model) (Van Dyne, 2010)	No	4 (factors)	independent
The Process of Cultural Competence in the Delivery of Healthcare Services (Campihna, 1998)	Yes	5 (constructs)	connected
Papadopoulos, Tilki and Taylor Model for Developing Cultural Competence / Intercultural Education of Nurses in Europe (PTT/IENE Model) (Papadopoulos, 1998)	Yes	4 (stages)	cyclical
Pyramid Model of Intercultural Competence (Deardorff, 2006)	No	5 (levels)	pyramid
**Global Health Learning Models**			
Global Health Experience Learning Progression (GHELP)	Yes	3 * (stages)	progression

* plus external and internal factors.

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
