# Peer review of "Development of a Global Health Learning Progression (GHELP) Model"

_pharmacy, 2020, doi:10.3390/pharmacy9010002_

Round 1

Reviewer 1 Report

I was excited to read this paper as while I have seen a lot of work on global health education and electives in resource-limited settings I had not yet seen any dedicated to these experiences by pharmacists. In that regard, I was a little disappointed that the denominator of the participants being pharmacists, had little to do with the paper, and in fact, it's difficult to tell if their GHELP model would have been any different if it were teachers, firefighters, etc. doing the reflecting. This isn't necessarily a problem as a sound framework that could be applied across learners would be valuable, but the authors attempt to differentiate that this is exclusive to healthcare.

I found the methodology difficult to read and after two readings wasn't exactly sure what had been done. It seemed like a mixed methods study with some quantitative survey components (though none were reported in results) and then a mix of open ended and focus group methods but again, these weren't presented separately or concisely in the results. In the results I was expecting to see how many people participated in the various aspects of the study (response rates, amount of focus group attendees, etc) but was left wanting there. In the way it's presented, it's possible only one person responded. How is the reader to determine if these themes are universal without a little more clarity on how many people contributed and some demographics about participants?

I was also surprised that cultural sensitivity was used as the end result of the growth journey. In most academic ventures on studying culture in medicine, sensitivity is a lower tier level (aware that there are differences and being cautious not to offend/step on cultural toes). Cultural humility (awareness that one can never be culturally competent in another culture, and is able to see how our lived experiences impact decision making in ways that are not objectively right or wrong) is considered the highest rung, so to speak. 

Another missed opportunity was the fact that there was no attempt to differentiate themes of comments from those working in low resource settings vs high resource settings. This dataset is unique in that the authors say they had a mix of both. Would be powerful to see how perspectives differed.

In short, the paper didn't make it clear to me that the methodology informed the GHELP model, but rather read as if the authors had a thesis and simply presented it against the backdrop of having done focus groups and surveys. We need more of the methodology and more meat in the results to be able to agree with the framework. 

Reviewer 2 Report

Whilst the paper as presented is of interest, I would have a few suggestions that the authors might like to take up in order to improve its potential impact.

  1. The authors use 2 basic methodologies: survey and discussion groups. However, the numbers involved are not the same. Why were there only 22 in the discussion group? Was there a selection of 22 from 81? And if so on what basis? Is there any possibility that the 22 may be biaised in some way by the process of selection?
  2. Substantial data is given for the survey approach but only rather "anecdotal" data for the discussion approach. Did the authors envisage using automatic content analysis software such as Leximancer?
  3. Sixteen countries were "chosen" - but on what basis? Simply on income? Was this choice based on actual data for previous foreign placements?
  4. The placements were apparently of variable duration - it would be useful to have some data on this point.
  5. Some references are given in full, e.g. for MAXQDA, but not for others, e.g. Qualtrics.
  6. The authors state that the GHELP still has to be validated? One might think therefore that this paper is somewhat premature?
  7. The authors received no external funding - so how was the project funded (on the bais of staff hours?)?

Round 2

Reviewer 1 Report

Paper is improved and new methods help clarify the addition to prior work the authors did in this space. It smells a little of salami slicing (https://www.editage.com/insights/the-pitfalls-of-salami-slicing-focus-on-quality-and-not-quantity-of-publications#:~:text=What%20is%20salami%20slicing%3F,article%20into%20smaller%20published%20articles.) and I'll leave to the editor to determine if that is ok.